# Solution Combustion Synthesis of Hafnium-Doped Indium Oxide Thin Films for Transparent Conductors

**DOI:** 10.3390/nano12132167

**Published:** 2022-06-23

**Authors:** Rita Firmino, Emanuel Carlos, Joana Vaz Pinto, Jonas Deuermeier, Rodrigo Martins, Elvira Fortunato, Pedro Barquinha, Rita Branquinho

**Affiliations:** CENIMAT|i3N, Department of Materials Science and CEMOP/UNINOVA, NOVA School of Science and Technology, NOVA University of Lisbon, 2829-516 Caparica, Portugal; r.firmino@campus.fct.unl.pt (R.F.); jdvp@fct.unl.pt (J.V.P.); j.deuermeier@fct.unl.pt (J.D.); rfpm@fct.unl.pt (R.M.); emf@fct.unl.pt (E.F.)

**Keywords:** transparent conducting oxide (TCO), solution combustion synthesis, indium oxide, hafnium dopant, rapid thermal annealing (RTA)

## Abstract

Indium oxide (In_2_O_3_)-based transparent conducting oxides (TCOs) have been widely used and studied for a variety of applications, such as optoelectronic devices. However, some of the more promising dopants (zirconium, hafnium, and tantalum) for this oxide have not received much attention, as studies have mainly focused on tin and zinc, and even fewer have been explored by solution processes. This work focuses on developing solution-combustion-processed hafnium (Hf)-doped In_2_O_3_ thin films and evaluating different annealing parameters on TCO’s properties using a low environmental impact solvent. Optimized TCOs were achieved for 0.5 M% Hf-doped In_2_O_3_ when produced at 400 °C, showing high transparency in the visible range of the spectrum, a bulk resistivity of 5.73 × 10^−2^ Ω.cm, a mobility of 6.65 cm^2^/V.s, and a carrier concentration of 1.72 × 10^19^ cm^−3^. Then, these results were improved by using rapid thermal annealing (RTA) for 10 min at 600 °C, reaching a bulk resistivity of 3.95 × 10 ^−3^ Ω.cm, a mobility of 21 cm^2^/V.s, and a carrier concentration of 7.98 × 10^19^ cm^−3^, in air. The present work brings solution-based TCOs a step closer to low-cost optoelectronic applications.

## 1. Introduction

Transparent conducting oxides (TCOs) are materials known to have low resistivity (ρ > 10^−4^ S cm^−1^) and high transparency (T_380–750nm_ > 80%) in the visible region of the spectrum, which requires a minimum bandgap of 3 eV [1,2,3,4,5,6]. Nevertheless these two properties are not usually linked together [2,6]. Historically, TCOs have been dominated by tin oxide (SnO_2_), indium oxide (In_2_O_3_), indium tin oxide (In_2_O_3_: Sn or ITO), and zinc oxide (ZnO) [4]. Their properties make these ideal for various applications such as transparent electrodes in solar cells [7,8], electroluminescent devices [9], transparent thin-film transistors (TFTs) [10], electrochromic devices [11], liquid crystal displays (LCD) [12] and organic light emission diodes (OLED) [9,13].

TCOs are usually prepared with thin-film technologies to allow application in different device architectures [1,3]. Most often, conventional high-vacuum techniques are used to produce TCO thin films due to their ability to yield dense films with high quality properties (e.g., low bulk resistivity and high carrier concentration), however, alternative TCO processing methods, such as solution synthesis, have garnered interest [1,4]. In a solution-based method, an oxide precursor solution is prepared by simply dissolving metal alkoxides or metal salts in an appropriate solvent, which can then be deposited through various techniques, such as spin coating [7,14,15], dip coating [16], spray pyrolysis [17,18], and inkjet printing [19], among others, to produce metal oxide thin films. However, typically highly toxic solvents such as 2-methoxyethanol and acetonitrile are used [20], and to guarantee the removal of impurities and the formation of metal–oxygen bonds, high processing temperatures are required. New advancements in solution processing, such as solution combustion synthesis (SCS), have allowed for a reduction in the temperature required to form the metal-oxygen-metal (MOM) framework and ensure promising conductivity [1].

SCS has been applied to produce diverse morphologies of metal oxides such as powder, nanostructures, and thin films [21,22]. For combustion synthesis to occur, three elements are necessary: a fuel, an oxidizer, and a high enough temperature to achieve combustion ignition and convert precursors to oxides [22]. Throughout the years, SCS has been reported to be a viable technique for the production of TCOs, as the films produced are highly homogenous and pure [21,23,24]. This leads to an enhancement of the mobility (µ), and consequentially, of the films’ conductivity [4].

TCOs typically exhibit low resistivity (ρ), below 10^−4^ Ω.cm, due to the tunable defect chemistry of metal oxides [1,25]. Increasing the conductivity can be achieved either by enhancing carrier concentration or by enhancing mobility of the thin films [14,25]. However, TCOs with very high carrier concentration result in a loss of film transparency. Therefore, to prevent undesired optical absorption, the maximum carrier concentration must be below 10^21^ cm^−3^ [2,14]. On the other hand, increased mobility can be achieved by increasing the scattering time or by decreasing the carrier effective mass [4]. The scattering time is related to film quality, whereas the decrease of the carrier effective mass (m^*^) is related to the development of new materials, such as doping with transition metal (TM) elements [4,15,26].

In intrinsic TCOs, when no dopant is added, the conductivity is due to oxygen vacancies or interstitial metal cations. In doped TCOs, free carriers are created due to the introduction of metal ions with different valences in the crystalline network [5]. This allows improved tuneability over optical and electrical properties and enhances the environmental stability of TCO thin films when compared to intrinsic TCO films [27]. Typically, extrinsic doping is achieved by replacing the host ion sites with same-period, next-row elements, e.g., Sn^4+^ to substitute In^3+^, to minimize disturbance of host crystal and electronic structure [27]. Nevertheless, other elements have been used as external impurities for In_2_O_3_, such as Mo, Ti, Zr, W, Ga, Al, B, F, Mg, Sb, and Ce [15].

Sn is the most-used dopant in In_2_O_3,_ as 60% of transparent conductors in the market are based on indium–tin oxide (ITO) [6,25,28,29]. In ITO, Sn^4+^ substitutes In^3+^ in the indium oxide crystal structure, resulting in the donation of one free electron [5,14,29,30]. Therefore, high densities of Sn are necessary in order to achieve high carrier concentrations. Moreover, the strong hybridization between Sn 5s-orbitals with In 5s-orbitals enlarges the m^*^ around the conduction band minimum (CBM), reducing µ [29]. That could be explained by the presence of resonant donor levels within the conduction band [26,29]. Xu et al [26]. investigated transition metal doping by simulation, concluding that Zr, Hf, and Ta could be better donors than Sn. Doping with Zr, Hf, and Ta leads to lower m^*^ due to d-orbital defect levels being of sufficiently high energy to allow the full ionization of the dopant without changing the dispersion around the CBM of In_2_O_3_.

Doped In_2_O_3_ films are usually polycrystalline, however, different dopants, dopant quantities, and/or process conditions result in distinct preferred orientations [31,32]. Substitutionally doping metal oxides with similar ionic radii elements can lead to little lattice distortion as the dopant substitutes in the host ion sites. The work of Kaleemulla et al. [31] corroborates this assumption. When In_2_O_3_ thin films were doped with Mo atoms, the lattice parameter was not significantly changed, as the ionic radius of Mo^6+^ (0.65 Å) is smaller than that of In^3+^ (0.80 Å) [33]. Similar behavior is expected for doping with Hf^4+^ (0.71 Å), Ta^5+^ (0.64 Å), and Zr^4+^ (0.72 Å), as they all have smaller, though rather similar, ionic radii than In^+3^ (0.80 Å) [33].

Typically, a dopant with a higher valence than the host ion that occupies the lattice sites replaces the host ion and donates free charge carries. This results in the Moss–Burstein shift, increasing the optical bandgap [31]. This was observed by Kaleemulla et al. [31] when doping In_2_O_3_ with Mo (1–3 at%). However, for higher doping (5 at%), the bandgap decreased, probably due to lattice distortion. Doping with Hf^4+^, Ta^5+^, and Zr^4+^ may lead to an increase of the bandgap, as the respective oxides have higher bandgap than In_2_O_3_, HfO_2_ (5.25–5.90 eV) [34,35], Ta_2_O_5_ (3.8–5.3 eV) [36], and ZrO_2_ (4.3–5.5 eV) [37,38,39]. Solution-based TCOs have been reported using different dopants as detailed in Table 1.

This work reports on the solution processing of transition-metal-doped In_2_O_3_ thin films, where different dopant quantities and annealing parameters are investigated in order to optimize the TCOs electrical properties. From the three most promising transition metals listed above, it was decided to use Hf as the dopant, considering the promising results achieved by Wang et al. [34]. The precursor solutions were prepared using 1-methoxypropanol as the solvent, due to its lower environmental impact compared to the widely-used and carcinogenic 2-methoxyethanol. TCO thin films were deposited by a simple spin-coating process where 100–110 nm thick films were achieved after 8-layers deposition. The more promising results are found to be the films doped with 0.5 M% Hf annealed in RTA in air. This simple, fast-annealing method produced films with superior opto-electrical properties than other solution-processed films, as can be seen from Table 1. The 0.5 M% Hf-doped In_2_O_3_ films achieved good electrical properties without compromising the transparency of the films.

## 2. Materials and Methods

### 2.1. Precursor Solutions Preparation and Characterization

The metallic oxide precursor solutions were prepared by dissolving a 0.2 M concentration of indium (III) nitrate hydrate (In(NO_3_)_3_·xH_2_O, Sigma, 99.9%, CAS 207398-97-8), in 1-methoxy-2-propanol (1-MP, C_4_H_10_O_2_, Roth, ≥ 99% CAS 107-98-2). For the combustion reaction, urea (CO(NH_2_)_2_, Sigma, 98%, CAS 57-13-6) was added as fuel to each precursor solution, with molar ratio between urea and indium nitrate of 2.5:1, to guarantee the redox stoichiometry of the reaction. After that, hafnium (IV) chloride (Hf Cl_4_, Alfa Easer, 98% CAS 13499-05-3) was used as a dopant source, in a range of 0.25–1 M%. For the 0.75 and 1 M% Hf solution 5% *v/v* of ethylene glycol (EG, CH_2_OHCH_2_OH, Carlo Erba, ≥99.5% CAS 107-21-1) was added to guarantee full dissolution of reagents.

All precursor solutions were stirred at 430 rpm in air environment at room temperature for at least 1 h, and were filtrated through 0.45 μm hydrophobic polytetrafluoroethylene (PTFE) filters.

The solutions were chemically and thermally characterized by Fourier Transform-Infrared Spectroscopy (FTIR), and by thermogravimetry and differential scanning calorimetry (TG DSC), respectively. FTIR spectroscopy characterization of the solutions was performed using an Attenuated Total Reflectance (ATR) sampling accessory (Smart iTR) equipped with a single bounce diamond crystal on a Nicolet 6700 FT-IR Spectrometer (Thermo Fisher Scientific, Waltham, MA, USA). The spectra were acquired with a 45° incident angle in the range of 4500–525 cm^−1^ and with a 2 cm^−1^ resolution. TG-DSC analysis were performed under air atmosphere up to 550 °C with a 10 °C/min heating rate in an aluminum crucible with a punctured lid using a simultaneous thermal analyzer, Netzsch (TG-DSC-STA 449 F3 Jupiter, Selb, Germany).

### 2.2. Thin Film Deposition and Characterization

Prior to deposition, all substrates (silicon wafer and Corning Eagle glass with an area of 2.5 × 2.5 cm^2^) were cleaned in an ultrasonic bath at 60 °C in acetone for 15 min, then in 2-isopropanol for 15 min. Then, the substrates were cleaned with deionized water and dried under N_2_; followed by a 15 min UV/Ozone surface activation step with a substrate-to-lamp distance of 5 cm, using a PSD-UV Novascan system (Ames, IA, USA).

The deposition of 8 layers of each precursor solutions was performed by spin coating for 35 s at 3000 rpm (Laurell Technologies, North Wales, PA, USA) on Corning glass substrates. Between layers a 5 min annealing was done on a hot plate at 400 °C, and a final annealing was performed for 1 h at the same temperature in ambient conditions. Then, In_2_O_3_ thin films doped with 0, 0.25, 0.5, 0.75, and 1 M% Hf were characterized.

The effect of rapid thermal annealing (RTA) on the TCO’s properties was studied by annealing the films stack at 600 °C for 10 min in air and N_2_ environment conditions, directly after the previous hot plate annealing at 400 °C. The RTA temperature was selected based on previous results as reported by Ullah et al. [45].

The relative humidity during the deposition of the films varied from 23.5 to 47%. After the thin film deposition, aluminum electrodes were deposited by thermal evaporation using homemade equipment via shadow mask in a Van der Pauw geometry to allow Hall effect measurements.

Films thickness was obtained by spectroscopic ellipsometry, made over an energy range of 1.5–5 eV with an incident angle of 70° using a Jobin Yvon Uvisel system (Chilly-Mazzarin, France) on films deposited on silicon wafers. The acquired data were modulated using the DELTAPSI software (v2.6.6.212, Horiba, Bensheim, Germany), and the fitting procedure was done pursuing the minimization of the error function (χ^2^). FTIR-ATR spectroscopy characterization of thin films deposited on Si substrates was performed as described for precursor solutions. The optical properties were obtained using a Perkin Elmer lambda 950 UV/VIS/NIR spectrophotometer (Llantrisant, UK). Transmittance (T%) was obtained from 200 to 2000 nm with a 3 nm step.

The structural analysis of the films deposited on Si was assessed by X-ray diffraction) using an MPD X’Pert PRO from PANalytical (Royston, UK) powder diffractometer, with a Cu Kα radiation source (λ = 1.540598 Å) and equipped with an 1D X’Celerator detector. The XRD measurements were performed in the range of 10 to 65° (2θ) in the Bragg-Brentano configuration, with a scanning step size of 0.0668° in 2θ.

The surface morphology and elemental analysis was studied by scanning electron microscopy (SEM, Zeiss Auriga Crossbeam electron microscope, Oberkochen, Germany) equipped with energy dispersive X-ray spectroscopy (EDS, Oxford X-Max150). Chemical composition was also assessed by X-ray photoelectron spectroscopy (XPS) using monochromated Al Ka irradiation (1486.6 eV) with a Kratos Axis Supra (Manchester, UK).

## 3. Results and Discussion

The influence of Hf doping on the electrical properties of In_2_O_3_ thin films was studied by preparing precursor solutions with a 0.2 M concentration of 1-methoxy-2-propanol (1-MP). Undoped In_2_O_3_ thin films were obtained from indium oxide precursor and Hf-doped In_2_O_3_ thin films were obtained by adding HfCl_4_ to the indium oxide precursor solution in the range of 0.25–1 M%.

### 3.1. Precursor Solutions Characterization

To understand the chemical bonds present in some of the precursor solutions, FTIR of solutions deposited on Si substrate was performed using ATR (Appendix A). As expected, most of the vibration bands in precursor solutions are attributed to the solvent, 1-methoxy-2-propanol (3500–2800 cm^−1^ and 1450–850 cm^−1^). Although the peak at 1110 cm^−1^ could be also attributed to Si-O bond from the substrate. Vibrations peaks attributed to C-O (1650 cm^−1^) and N-H (1575 cm^−1^) are typical of urea [46], and N-O vibration (1510 cm^−1^) is correlated with nitrates from the indium precursor.

The decomposition behavior of the metal oxide precursors was investigated by TG-DSC for undoped and Hf-doped In_2_O_3_ precursor solutions (Figure 1).

It is observed that both solutions have similar behaviors, showing only a minor shift to the right when the Hf precursor (HfCl_4_) is added. This was expected, as the combustion reaction temperature depends on the specific bonding energy of the ligands with the metal ions, which differs from metal to metal [47]. The undoped and the Hf-doped precursor solutions present an exothermic peak at 309 °C and 316 °C, respectively, with corresponding abrupt mass loss that can be assigned to the combustion reaction of the organic fuel with the metal nitrates. The endothermic peaks at 165 °C for the undoped precursor solution and at 168 °C for the Hf-doped solution can be attributed to the solvent evaporation and material condensation.

Thermal analysis of the precursor solutions indicate that the annealing temperature required for the full degradation of organics and to achieve precursor to oxide conversion is higher than 316 °C.

### 3.2. Doping of Indium Oxide Thin Films

Undoped and Hf-doped In_2_O_3_ thin films were produced by spin-coating precursor solutions and annealing at 400 °C to guarantee the complete formation of the metal oxide.

FTIR spectra of all films were obtained after annealing and compared with the spectra of precursor solutions to confirm precursor to oxide conversion. Figure 2 shows selected spectra of undoped and Hf-doped In_2_O_3_ thin films, before and after annealing (spectra of all thin film conditions is shown in Appendix A and Appendix A).

It is possible to confirm that the reaction occurred to its full extent as none of the organic absorbance peaks presented in the solution spectra are visible in the annealing films, Appendix A. The absorbance peaks that appear between 890 and 610 cm^−1^ are characteristic of vibrational modes of In-O bond [2].

The thickness of the doped thin films was accessed by spectroscopic ellipsometry for all conditions and is given in Table 2. All films have a thickness between 102 nm and 111 nm, with a slight increase in the film thickness for doping concentrations up to 0.5 M%, which then decreases with increased doping percentage (0.7 M% and 1 M%). This may be related to lower solubility of higher Hf% solution even with the addition of EG.

Figure 3a shows the transmittance spectra of undoped and Hf-doped thin films. All films have transmittance above 80% in the visible region of the spectrum, as presented on Table 2. The transmittance of doped indium oxide thin films sharply drops around 400 nm because of the In_2_O_3_ absorption edge, which is an indicator of good crystallinity [48].

The optical bandgap (E_opt_), determined via Tauc plot, increases when a dopant is added (Table 2). This can be the result of a shift in the Fermi level, known as the Moss–Burstein shift (Δ*E_BM_*), and could be caused by additional free carriers that were provided if the dopant ions occupied the lattice sites replacing some of the In ions [2,5,14,31]. Another reason for this behavior could be doping with a metallic ion for which oxide has a higher bandgap, as is the case of HfO_2_ (5.25–5.90 eV) [4,34,35].

In Figure 3b, the XRD patterns of In_2_O_3_ thin films doped with Hf are presented. All the diffracted peaks are attributed to the In_2_O_3_ cubic bixbyite structure (ICDD file no. 00-006-0416). This means that the dopant used, and its quantity, do not significantly affect the structure of the In_2_O_3_ thin films. Additionally, no change in the preferred orientation [31,49,50] nor any major shift in the angle of the XRD peaks were detected, which implies that Hf^4+^ ions substituted In^3+^ in the matrix at its regular lattice sites with minimal impact in the lattice parameter [31]. This should be related to the comparable, but slightly smaller, ionic radius of the dopant, Hf^4+^ (0.71 Å), compared to In^3+^ (0.80 Å) [33].

Hafnium incorporation in In_2_O_3_ thin films has improved films quality, as the diffraction peaks are more prominent and sharper. For higher molar percentages of Hf, the thin films show increased crystallite size, as shown in Figure 3c. The crystallite size (D) was determined using the Debye–Scherrer relation:(1)$$D=\frac{0.9\;\lambda }{B\;\cos\theta },$$
where λ is the X-ray wavelength (1.54 Å), B is the full width at half maximum (FWHM) of the Bragg diffraction angle 2θ. For 1 M% Hf, the solubility limit might have been reached, and some of the dopant might have been filtered, justifying the lower crystallinity and crystallite size, compared to the results of 0.75 M% Hf. This higher crystallinity can be explained by the presence of ethylene glycol (EG) in the precursor solutions of 0.75 and 1 M% thin films. Once EG acts as a chelating agent, it creates a homogeneous network with the metal ions, leading to an organized polymerization [22].

The study of the morphology of the doped In_2_O_3_ thin films on Si substrates was performed by SEM. Figure 3c inset shows the SEM images of the undoped and the 0.5 M% Hf-doped In_2_O_3_ thin films. The films present a similar smooth granular structure, indicating the polycrystallinity of the films (more details on the effect of annealing in the films’ morphology can be seen in Appendix A), which is in accordance with XRD results.

The elemental analysis of the doped In_2_O_3_ thin films was performed by energy-dispersive X-ray spectroscopy (EDS) (Figure 3d) and X-ray photoelectron spectroscopy (XPS) (Figure 4a,b and Appendix A and Appendix A). The EDS analysis indicates the presence of hafnium and that its quantity is similar to the target. Furthermore, it reveals that, aside from indium, oxygen, and the respective dopant, only carbon and silicon from the environment and substrate, respectively, were detected.

XPS was also performed to investigate the chemical bonding properties and stoichiometric information in Hf-doped thin films. Figure 4a,b show the high-resolution XPS spectra of Hf 4*d* (Hf 4*d*_5/2_ and Hf 4*d*_3/2_) and O 1*s*, respectively, for 0.5 M%-doped films. The spectrum for all conditions is presented in Appendix A. The binding energy of Hf 4*d*_5/2_ and Hf 4*d*_3/2_ is located at ~213.6 and ~224.2 eV, respectively. The existence of these two peaks indicates that Hf^4+^ is present in In_2_O_3_ thin films. The deconvoluted O 1*s* peak is very similar for all studied films (Appendix A). The observed peak at ~530.24 eV is associated with the indium oxide peak (In-O-In). The peak at ~531.18 eV is ascribed to O atoms in the vicinity of an oxygen vacancy (V_O_). The peak at ~532.17 eV is attributed to In-OH species on the surface. The last peak at ~533.16 eV can be assigned to molecularly chemisorbed H_2_O [51,52]. Appendix A shows the area percentage of these four peaks for all conditions.

The electrical properties of doped In_2_O_3_ thin films were measured by Hall effect and are presented in Figure 4c,d. It can be concluded that lower bulk resistivity is achieved with 0.5 M% Hf, with ρ of 5.73 × 10^−2^ Ω.cm. Regarding the film’s hall mobility, higher mobility was achieved for 0.5 M% Hf and not for the more crystalline films, 0.75 and 1 M% Hf. This could be due to ionized impurity scattering originated from a higher carrier concentration [32]. Regarding the carrier concentration (N), it is clear that, by increasing the doping level of Hf, N also increases. Considering that Hf^4+^ ions substitute at In^+3^ sites, it can be calculated that one free carrier is generated per substituted cation.

To summarize, the best electrical properties are achieved when indium oxide is doped with 0.5 M% Hf. For this reason, this condition was chosen for the study with rapid thermal annealing (RTA).

### 3.3. Effect of Post-Annealing in Undoped and Hf-Doped Indium Oxide Thin Films

In this section, another post deposition thermal treatment is studied in order to improve the electrical properties of TCOs in a short processing time. A post deposition treatment of 10 min in a rapid thermal annealing (RTA) system was performed in air and in N_2_. These processes have been applied before in literature, showing promising results [45,50,53].

After the deposition of the undoped and 0.5 M% Hf thin film, a final annealing at 600 °C was performed for 10 min in an RTA chamber. The films thickness decreased when RTA was performed, as expected due to enhanced film densification at higher temperatures. The undoped and the 0.5 M% Hf-doped films obtained thicknesses of ~79 nm and ~92 nm, respectively (Appendix A), corresponding to an average reduction in film thickness of 12 nm after annealing.

Regardless of the annealing conditions, doped thin films show slightly higher transmittances in the visible region (Appendix A). The average transmittance in the visible region (T_380–750nm_) of undoped In_2_O_3_ is ~78%, whereas for Hf-doped In_2_O_3_ thin films, T_380–750nm_ is ~82%, which can be related to the increased crystallinity of the doped thin films.

Considering the thin-films bandgap presented in Figure 5a, it is noticeable that the E_opt_ of doped thin films is higher than for undoped thin films for all annealing conditions. This can be justified by the Moss–Burstein effect and the high bandgap of HfO_2_, as mentioned previously. Additionally, it is possible to observe that the optical bandgap of the films decreases for annealing in RTA in air, and further for RTA in nitrogen, when compared to annealing on a hot plate at 400 °C, which is consistent with results reported in literature [54,55].

The XRD diffractograms of the TCOs with different annealing conditions (Appendix A) show that the new post-deposition conditions by RTA at 600 °C do not change the films’ structure, remaining cubic bixbyite. After RTA, the thin films continue to be polycrystalline, but crystallinity is improved, as the diffraction peaks become narrower. This result is expected since the use of higher temperatures leads to the increase of the level of crystallinity due to lattice relaxation [45]. The crystallite size has a drastic increase with the RTA post-deposition treatment, as presented in Figure 5b [45,56]. This is also observed in SEM images (in Figure 5b inset and Appendix A), where there is clear grain growth when RTA is applied. Additionally, the AFM analysis (Appendix A) revealed that both non-doped and doped thin films annealed in air present a small surface roughness of less than 1 nm, even after the RTA annealing.

To understand the effect of rapid thermal annealing on the composition of the thin films, EDS and XPS analysis were performed. The presence of Hf^4+^ in the films is attested by the EDS analysis for the 0.5 M% Hf-doped thin films (Appendix A). Nevertheless, it seems that the Hf^4+^ quantity present in these films is, in almost all cases, inferior to what was expected. The XPS analysis of the doped TCOs with different annealing conditions is presented in Appendix A. The XPS results after the annealing do not reveal any significant change due to the annealing (Appendix A).

The electrical properties of the TCOs with different post-deposition treatments are presented in Figure 6. When RTA is performed, the electrical properties of the thin films are clearly enhanced. As Hall mobility and carrier concentration increased, consequently, bulk resistivity decreased. Undoped thin films’ bulk resistivity decreased from 4.27 × 10^−1^ Ω.cm to 2.77 × 10^−2^ Ω.cm when RTA was performed in nitrogen. Additionally, the atmosphere of RTA annealing influenced the electrical properties, as the Hall mobility had higher values when the annealing was performed in air than in nitrogen, and the opposite occurred for the carrier concentration. For RTA in N_2,_ a lower mobility is observed when carrier concentration is higher, which can be attributed to ionized impurity scattering as well as to lower crystallinity, compared to RTA in air, resulting in increased grain scattering [32].

For the case of Hf-doped In_2_O_3_, the best annealing condition was RTA in air, as the ρ decreased from 5.73 × 10^−2^ Ω.cm to 3.95 × 10^−3^ Ω.cm. This improvement could be attributed to the higher crystallite size that allowed for the grain boundary scattering time to increase, as mobility reached values of 21 cm^2^/V.s [45]. The grain growth can also explain the carrier concentration enhancement, as annihilation of grain boundaries occurs. RTA annealing can also promote the diffusion of Hf atoms in In_2_O_3_ and the activation of Hf dopants, which leads to the improvement of the carrier concentration, to a value of 7.98 × 10^19^ cm^−3^ [56].

To study the stability of the thin films, the measurements were repeated after 3 weeks and, as can be seen in Figure 6, all electrical properties were improved. This slight enhancement can be attributed to fewer weakly bound oxygen species after the annealing treatments and more deep-level defects passivated with strong bonds [57,58,59,60,61,62].

The obtained results for 0.5 M% Hf-doped In_2_O_3_ thin films after RTA at 600 °C in air surpass the state-of-the-art of solution-processed TCOs, as can be seen in Table 1. A balance between good optical and electrical properties was achieved using a simple and fast process.

## 4. Conclusions

In summary, the importance of doping and post-deposition treatments was clearly demonstrated in solution-based indium oxide produced by solution combustion synthesis. The Hf-doped films showed transmittance above 80% and E_opt_ between 3.66 and 3.85 eV. Hall effect measurements demonstrated that the 0.5 M% Hf-doped thin film annealed at 400 °C showed the best electrical performance, reaching a bulk resistivity of 5.73 × 10^−2^ Ω.cm. These thin films improved their electrical performance after a post-deposition annealing in RTA for 10 min at 600 °C in air. A bulk resistivity of 3.95 × 10^−3^ Ω.cm was achieved, and these results surpass the state-of-the-art of Hf-doped TCOs and solution processed indium oxide TCOs, as depicted in Appendix A and Table 1, respectively. Compared to the solution-processed Hf-doped films [34], the present work sustains some advantages, such as smaller annealing time and higher mobility. These results clearly demonstrate that solution-based TCOs can be used in various optoelectronic applications, such as photovoltaic devices or transparent electronics.

## Figures and Tables

**Figure 1 nanomaterials-12-02167-f001:**
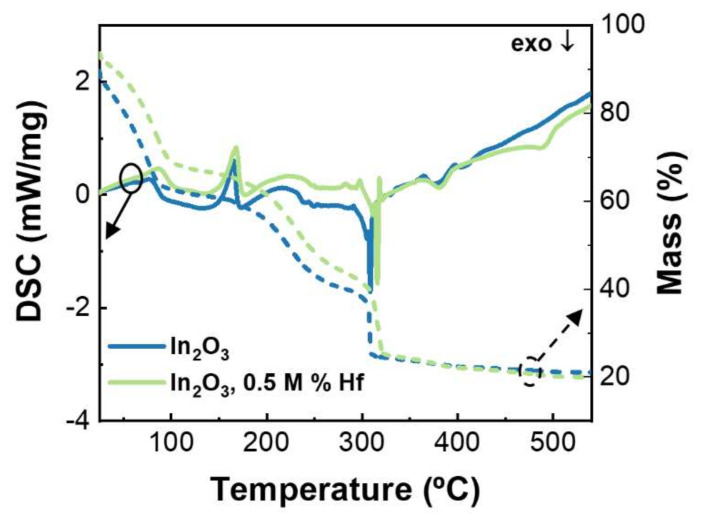
TG-DSC analysis of the undoped and 0.5 M% Hf-doped indium oxide precursor solutions.

**Figure 2 nanomaterials-12-02167-f002:**
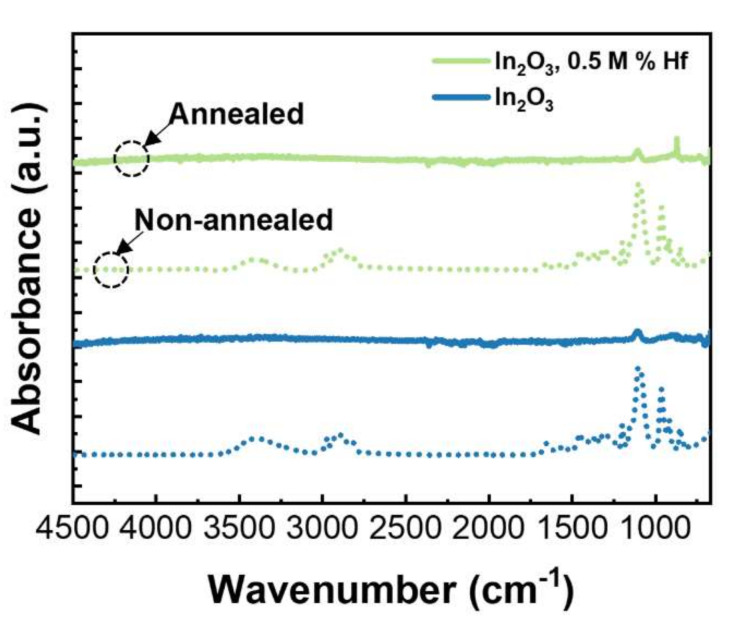
FTIR spectra of undoped and Hf-doped In_2_O_3_ thin films, before and after annealing.

**Figure 3 nanomaterials-12-02167-f003:**
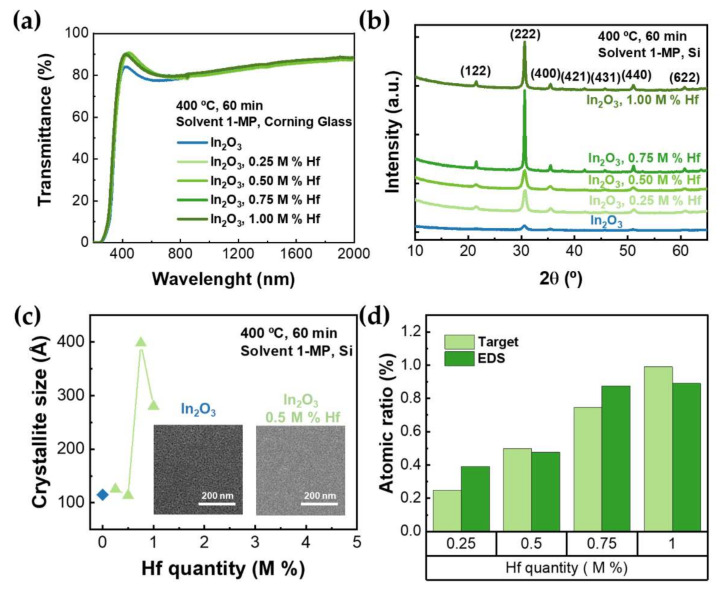
(**a**) Transmittance spectra of undoped and Hf-doped In_2_O_3_ thin films with different dopant quantities, annealed at 400 °C in air; (**b**) X-ray diffraction (XRD) of In_2_O_3_ films doped with different quantities of Hf; (**c**) crystallite size (Å) of undoped and Hf-doped In_2_O_3_ thin films; inset shows the scanning electron microscopy (SEM) of undoped and 0.5 M% Hf-doped In_2_O_3_ films; (**d**) atomic ratio (%) of Hf in In_2_O_3_ films obtained by electron dispersive X-ray spectroscopy (EDS) analysis and target atomic ratio for each dopant quantity.

**Figure 4 nanomaterials-12-02167-f004:**
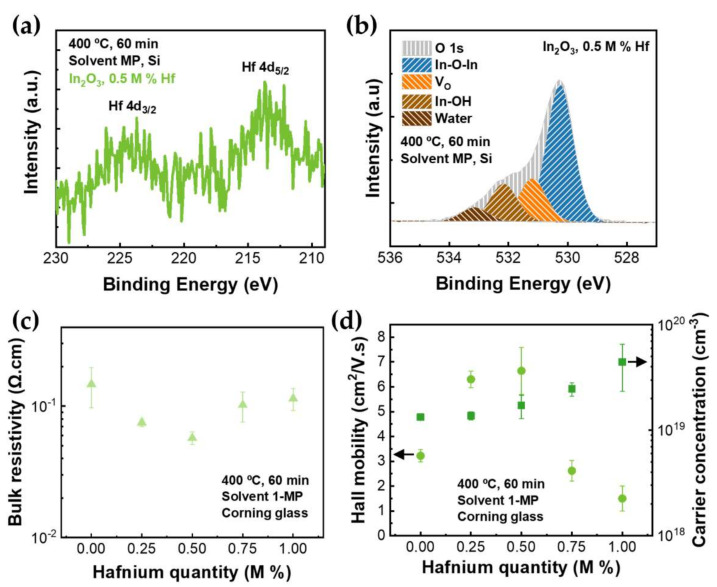
(**a**) XPS spectra of the Hf4*d* and (**b**) the deconvolution of O1s peak of films formed at 0.5 M% of Hf-doping levels; (**c**) bulk resistivity and (**d**) Hall mobility and carrier concentration obtained by Hall effect measurements for the undoped and Hf-doped In_2_O_3_ films.

**Figure 5 nanomaterials-12-02167-f005:**
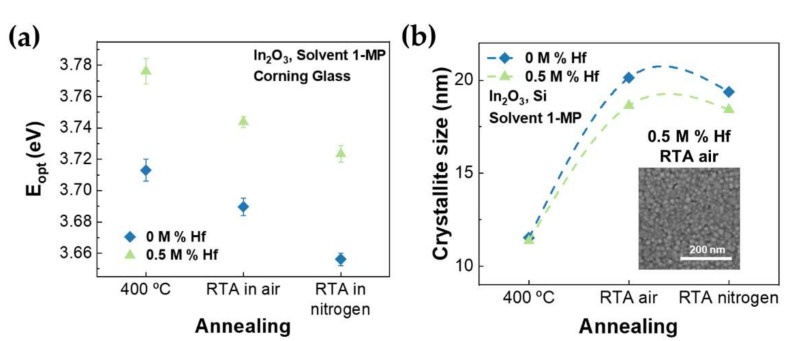
(**a**) Optical bandgap (eV) and (**b**) crystallite size (nm) of undoped and Hf-doped In_2_O_3_ thin films with different annealing conditions; inset shows the SEM surface image of 0.5 M% Hf-doped In_2_O_3_ rapid thermally annealed in air for 10 min at 600 °C.

**Figure 6 nanomaterials-12-02167-f006:**
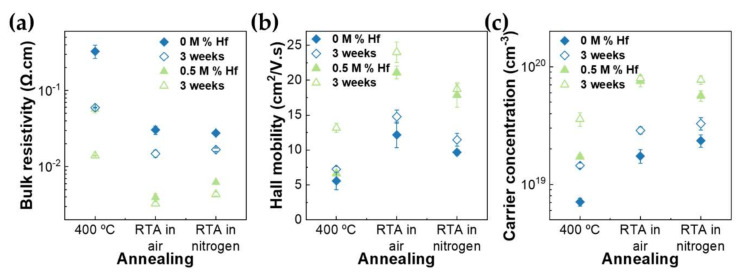
(**a**) Bulk resistivity (Ω.cm), (**b**) Hall mobility (cm^2^/V.s) and (**c**) carrier concentration (cm^−3^) of undoped and Hf-doped In_2_O_3_ thin films with distinct post-deposition treatments.

**Table 1 nanomaterials-12-02167-t001:** Electrical properties of transparent conducting oxides prepared by solution methods.

Year	TCO	T_annealing_ (°C)	Time of Annealing	T_550nm_ (%)	ρ (Ω.cm)	N (cm^−3^)	µ (cm^2^/V.s)
2003 [16]	Sn (6.1 at%)—In_2_O_3_	600	1 h	80	2.1 × 10^−4^	1.0 × 10^21^	28.0
2009 [40]	Mo (6 at%)—In_2_O_3_	400	-	75	8.1 × 10^−4^	1.9 × 10^20^	34.0
2015 [34]	Hf (3 at%)—ZnO	550	2 h	75	5.6 × 10^−3^	4.5 × 10^19^	2.5
2016 [41]	Zn (1 at%)—In_2_O_3_	250	2 h	80	2.8 × 10^−3^	2.3 × 10^20^	1.0
2016 [42]	F (2 at%)—In_2_O_3_	400	45 min	65	6.6 × 10^−4^	2.4 × 10^21^	11.3
2017 [43]	In_2-2x_Zn_x_Sn_x_O_3_; x = 0.3	400	30 min	83	1.0 × 10^−2^	6.7 × 10^19^	6.0
2018 [44]	Ti (0.3 M)—In_2_O_3_	450	15 min	97	5.0 × 10^−2^	1.0 × 10^19^	5.0
2020 [15]	W (0.5 at%)—In_2_O_3_	500	3 h	77	5.4 × 10^−4^	5.2 × 10^20^	23.0
**This work**	Hf (0.5 M%)—In_2_O_3_	600	10 min	80	4.0 × 10^−3^	8.0 × 10^19^	21.0

**Table 2 nanomaterials-12-02167-t002:** Thickness, transmittance on the visible region and optical bandgap of Hf-doped indium oxide thin films.

Hf (M%)	Thickness (nm)	T_380–750nm_ (%)	E_opt_ (eV)
0	104.0 ± 0.6 nm	79.45 ± 2.26	3.72 ± 0.01
0.25	107.9 ± 1.2 nm	83.33 ± 3.88	3.77 ± 0.01
0.50	110.9 ± 1.3 nm	83.85 ± 4.06	3.78 ± 0.01
0.75	106.8 ± 0.5 nm	83.53 ± 3.66	3.85 ± 0.01
1.00	101.8 ± 0.4 nm	83.16 ± 3.63	3.85 ± 0.01

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
