# Peer review of "Solution Combustion Synthesis of Hafnium-Doped Indium Oxide Thin Films for Transparent Conductors"

_nanomaterials, 2022, doi:10.3390/nano12132167_

Round 1

Reviewer 1 Report

The authors of this paper reported the fabrication of high-transparency Hf-doped indium oxide thin films by controlling the Hf dopping concentration, and post annealing temperature. As a result, optimised oxide films with a bulk resistivity of 3.95 × E^-3 Ω.cm, a mobility of 21 cm^2 /Vs and a carrier concentration of 7.98 × E^19 cm^- 3 were obtained. The results are encouraging. This paper can be published, assuming the author can further clarify the following questions that the reviewers raised.

1. Why is the rapid annealing temperature (RTA)  chosed to be 600c?  how about other temperature? how does the film performance depend on the RTA?

2. Fig.3c and Fig.S3 presented the surface morphology of the dopped oxide film by SEM characterizaiton. Maybe AFM analysis is better in terms of the surface roughness (RMS) characerization. The authors may comments on the effect of RMS on the transparency of the dopped oxide film.

3. Fig.4b suggested C and H atom were introduced into the oxide film. Please comments their possible effect on the film quality.  Although the author mentioned their effects at line 347, but no further explaination was given.

line 347: "Also, it seems that films with higher percentages of hydroxides and metal carbonates or water reveled to have best electrical performance" 

4. "This higher crystallinity can be explained by the presence of ethylene glycol (EG) in the precursor solutions of 0.75 and 1 M % thin films"--- The authors should provide some experiment data to justify this argument. 

5.  line 190, and line 268.  figures are not correctly referenced. please correct them

Reviewer 2 Report

In the manuscript „Solution combustion synthesis of hafnium-doped indium oxide thin films for transparent conductors” the authors present a complex study of hafnium-doped indium oxide layers for applications as TCO.

            The manuscript is well written and different samples are prepared and characterized in order to find the most appropriate experimental conditions to obtain layers with adequate optical and electrical properties.

For developing TCO layers, a solution based method is presented and a low environmental impact solvent is used. Even the electrical resistivity value (a vital requirement for the TCO) is far from those recorded for films obtained by other deposition techniques (for example sputtering, reference 36 in text or reference 8 in supplementary material), the films have transmittance above 80 % in the visible region of the spectrum.

The Introduction part can be enlarged with the description of the deposition methods used in the fabrications of the best TCO films (optical and electrical).

In the Table S5, the T380-750nm (%) values can be inserted, as in the Table 2.

Round 2

Reviewer 1 Report

With these modifications, I think the paper can be accepted.